environmental science

bioretention system, microbial fuel cell, electrode spacing, wastewater treatment, electricity production efficiency

**Authors for correspondence:**
Wang Ya-Jun
e-mail: wyj79626@163.com
ZhaoYang Wang
e-mail: wangzhaoyanghit@126.com

# The influence of electrode spacing on the performance of bioretention cell coupled with MFC

Wang Ya-Jun[1], Chen Tian-Jing[1], Li Jin-Shou[1], Si Yun-Mei[2] and ZhaoYang Wang[2]

[1]School of Civil Engineering, Lanzhou University of Technology, 287 Langongping, Lanzhou 730050, People's Republic of China
[2]College of Earth and Environmental Science, Lanzhou University, Lanzhou 730000, People's Republic of China

(iD) ZYW, 0000-0003-0678-0346

In order to explore the influence of electrode spacing on the performance of the enhanced bioretention system, four bioretention cells with microbial fuel cell (BRC–MFC) systems with different electrode spacing were designed, and the effect of electrode spacing on system performance was revealed by analysing its water treatment capacity and electricity production efficiency. The results showed that BRC–MFC had good water treatment capacity and could produce electricity simultaneously. Compared with other BRC–MFC systems with spacing, the BRC3 system (with an electrode spacing of 30 cm) had significant water treatment capacity under different organic loads, especially under high organic load (C/N = 10) operation, COD removal rate was as high as 98.49%, $NH_4^+ - N$ removal rate was as high as 97%, and it had a higher output voltage of $170.46 \pm 6.17$ mV. It could be seen that proper electrode spacing can effectively improve the water treatment capacity of the BRC–MFC system. This study provided a feasible method for improving the performance of the BRC–MFC system, and revealed the relevant mechanism. A proper electrode spacing with sufficient carbon sources could effectively improve the water treatment capacity of the BRC–MFC system.

## 1. Introduction

Regarding water pollution, the world is facing two important challenges: one is compound pollutant pollution, and the other is energy shortage [1,2]. In order to solve these two problems at the same time, a sustainable wastewater treatment technology has gradually become a research hotspot.

Traditional wastewater treatment plants rely on the combined effects of physics, chemistry and biology. This process requires mechanical equipment and a large amount of energy input to achieve the removal of pollutants. However, bioretention cells (BRCs) are similar to constructed wetlands, which is a kind of ecological treatment system, using physical, chemical and biological effects to achieve removal effect, imitating the natural environment, and have good wastewater treatment effects without the need for energy input. Studies had shown that BRC has a significant removal effect on pollutants such as TSS [3], COD [4,5] and heavy metals [6]. Due to its small size, simple structure, good treatment effect, low cost and strong sustainability, it has become a potential wastewater treatment technology [7]. At present, the research on BRC has mainly focused on the operation efficiency of facilities and the optimization of design parameters (such as matrix type, gradation, hydraulic load, height of submerged layer, etc.) [8–10].

The biogeobattery reflected a natural phenomenon that occurred at the oxidation–reduction interface of the earth's surface. It used electrons generated by microorganisms oxidizing organic carbon, sulfide and other electron donors in the anaerobic area were transmitted to the aerobic area through a 'long distance' through the extracellular mediator, and then underwent a reduction reaction with electron acceptor such as oxygen [11]. However, the operating principle of microbial fuel cells (MFCs) is similar to this. In recent years, as an emerging technology, it has great potential for simultaneous wastewater treatment and electricity generation, and it has attracted widespread attention from scholars [12]. MFC uses wastewater pollutants as fuels and converts these into electrical energy through electrochemically active microbial degradation compounds [13], making MFC a sustainable technology. In order to effectively play the MFC function, the anode area must be kept in an anaerobic state, while the oxygen in the cathode area combines with protons and electrons to form a circuit. The reactions that generally occur in MFC can be summarized by the following equations [14]:

$$\text{Anodic reaction: } CH_3COO^- + H_2O \rightarrow 2CO_2 + 2H^+ + 8e^-, \tag{1.1}$$

$$\text{Cathodic reaction: } O_2 + 4e^- + 4H^+ \rightarrow 2H_2O, \tag{1.2}$$

and
$$\text{Overall reaction: } C_6H_{12}O_6 + 2H_2O + 6O_2 \rightarrow 6CO_2 + 8H_2O. \tag{1.3}$$

There had been some early studies on independent BRC systems and MFC systems [15,16]. However, the coupling of BRC and MFC is still in the initial stage of research. In this study, BRC and MFC were coupled to form a BRC–MFC system, in which the entire reaction column had both anaerobic and aerobic conditions. When the organic matter in the anode area was oxidized, it would produce the electrons ($e^-$) and protons ($H^+$), then moved to the cathode, and the external circuit transferred the electrons from the anode to the cathode through the insulated copper wire to generate current [17]. By studying the effect of electrode spacing on the performance of BRC–MFC, on the one hand, this study could test the feasibility of the coupling system of BRC and MFC to treat wastewater and generate electricity at the same time, which laid the foundation for future research, and on the other hand to explore the influence of electrode spacing on the water treatment capacity of the BRC–MFC system, and to obtain the best electrode spacing, which provided a reference for future practical application.

# 2. Material and methods

## 2.1. Experimental material

The BRC–MFC column was constructed with non-opaque Perspex, which was 800 mm in height and 80 mm in inner diameter. In the BRC system, the composition and filling of fillers follows the Australian FAWB adoption guidelines [18]: from top to bottom were super high-rise (100 mm), filter layer (300 mm, filled with fine sand with a diameter of 0.15–1.00 mm), transition layer (100 mm, filled with coarse sand with a diameter of 0.50–1.00 mm), submerged layer (200 mm, filled with medium sand with a diameter of 0.25–0.50 mm) and drainage layer (100 mm, filled with gravel with a diameter of 0.15–1.00 mm). The focus of this study was to examine the removal effect by changing the electrode spacing inside the BRC–MFC, so no plants were planted. The BRC–MFC column had a high water outlet to keep the submerged area in an anaerobic state, and added 5% pine wood chips as a supplementary carbon source in the submerged area. The BRC–MFC column used granular activated carbon (GAC) as the electrode materials, and the volume of the anode and the cathode were both 251.20 cm$^3$. Insulated copper wire was used internally to connect the anode and cathode, and

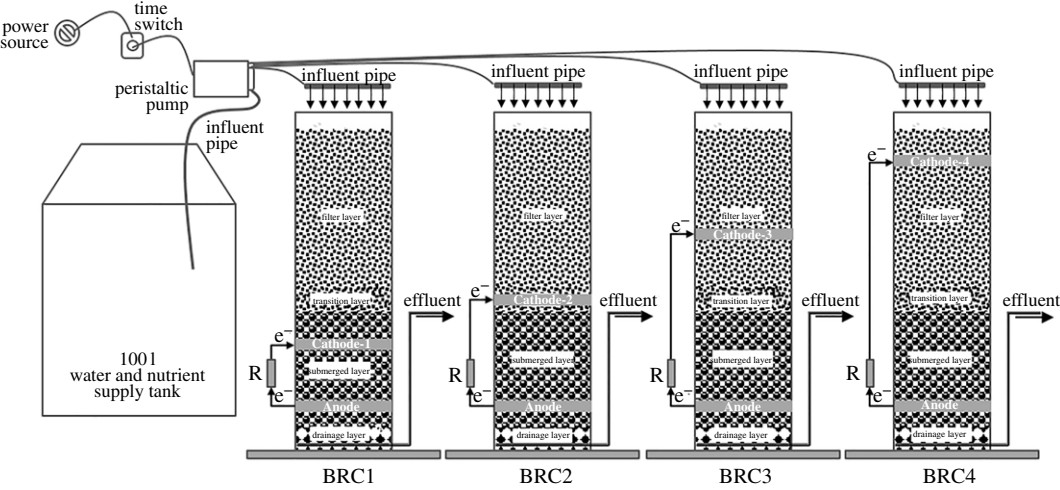

**Figure 1.** Schematic diagram of BRC–MFC optimal electrode spacing test device.

connected externally to 1000 Ω resistors to complete the circuit. In order to study the influence of electrode spacing on BRC–MFC processing efficiency and electricity generation performance, four identical BRC–MFC systems were designed. The reaction columns were shown in figure 1. The anode was located 15 cm from the bottom of the reaction column, and the distance between the anode and the cathode was changed by adjusting the position of the cathode layer (BRC1 system electrode spacing 10 cm; BRC2 system electrode spacing 20 cm; BRC3 system electrode spacing 30 cm; BRC4 system electrode spacing 40 cm).

## 2.2. Experimental procedure

A multi-channel peristaltic pump (WT600-2 J, Longer, China) was used for intermittent water intake, with a hydraulic load of $1.0 \, m^3/(m^2 \, d)$, and the treated wastewater was discharged from the drain and collected. There were four stages in the experiment: (i) the first stage, C/N = 2 : 1; (ii) the second stage, C/N = 4 : 1; (iii) the third stage, C/N = 8 : 1; (iv) the fourth stage, C/N = 10 : 1. Each reaction column was tested in triplicate.

## 2.3. Inoculation and synthetic wastewater

The activated sludge was taken from the anaerobic reactor of the Lanzhou Wastewater Treatment Plant in China as the submerged layer and GAC inoculated sludge in the reaction column. Before the experiment was started, the inoculated sludge was domesticated and cultivated for 45 d.

In order to reduce the difficulty of analysis mechanism and detection caused by the complex water quality and fluctuating water volume of actual domestic sewage, during the research process, artificial water distribution was used to simulate actual domestic sewage as the source of influent, and add different amounts of glucose as carbon source to synthetic wastewater. The main components of synthetic wastewater were shown in table 1 (all reagents are of analytical quality).

## 2.4. Experimental determination methods

COD was determined with an HACH DR2800 (DR2800, Hach, USA). $NH_4^+ - N$ was measured using a spectrophotometer UV-1800 (Shimadzu Corp., Japan). All the above parameters were determined by the methods and procedures described in 'Water and Wastewater Monitoring and Analysis Methods' [19].

The voltage was monitored and collected by a Midi LOGGER GL820 data acquisition instrument (GL820, Japan Graphic Technology Co., Ltd., Japan) in real time, and the electric potential was collected by a VC890D multimeter (VC890D, Shenzhen Yisheng Shengli Technology Co., Ltd., China). The output voltage $E$ (mV) was automatically measured every 1 s; the electrode potential was

**Table 1.** Synthetic wastewater composition.

| reagents | reagent content (g/100 l) | reagents | reagent content (g/100 l) |
|---|---|---|---|
| glucose | 13.988(C/N = 2 : 1) | $NiCl_2 \cdot 6H_2O$ | 0.1967 |
|  | 24.988(C/N = 4 : 1) |  |  |
|  | 49.363(C/N = 8 : 1) |  |  |
|  | 79.300(C/N = 10 : 1 |  |  |
| $NH_4Cl$ | 22.2402 | $MnCl_2 \cdot 4H_2O$ | 0.0058 |
| $K_2HPO_4$ | 2.1777 | $CuSO_4$ | 0.0058 |
| $NaHCO_3$ | 8.73 | $ZnCl_2$ | 0.0058 |
| $FeCl_2 \cdot 4H_2O$ | 4.8151 | $CaCl_2$ | 0.0427 |
| $CoCl_2 \cdot 6H_2O$ | 0.3913 | Humic acid | 0.1213 |

measured using Ag/AgCl electrode as a reference electrode. Current ($I = V/R$) and power ($P = VI$) were determined by basic electrical calculations.

Total volume power density $P_d$ (mW m$^{-3}$):

$$P_d = \frac{E^2}{V \cdot R_{ext}}. \tag{2.1}$$

Where $P_d$ is the total volume power density (mW m$^{-3}$), $V$ is the effective volume of the reaction column packing (m$^3$), this study is $3.52 \times 10^{-3}$ m$^3$. (Although the electrons come from the anode, some researchers also used the anode volume to unitize the output power [16], but the anode and cathode electrode regions contribute to the total volume of the reaction column. Therefore, this experiment used the effective volume of the reaction column packing to calculate the total volume power density $P_d$).

Anode volume power density $P_a$ (mW m$^{-3}$):

$$P_a = \frac{E^2}{V_a \cdot R_{ext}}. \tag{2.2}$$

Where $P_a$ is the total volume power density (mW m$^{-3}$), $V_a$ is the anode effective volume (m$^3$), this study was $0.2512 \times 10^{-3}$ m$^3$. (In order to facilitate comparison with related literature data, the effective volume of the anode was introduced to calculate the anode volume power density $P_a$.)

Coulombic efficiency CE (%) was calculated using the formula as follows:

$$CE = \frac{M \cdot I}{F \cdot q \cdot n \cdot \Delta COD}. \tag{2.3}$$

Where $M$ is the molecular weight of oxygen (32 g mol$^{-1}$); $F$ is the Faraday's constant (96 485 C mol$^{-1}$); $q$ is the water flow rate (l s$^{-1}$); $n$ is the number of electrons per mole of oxygen (4 mol e$^-$/mol O$_2$); $\Delta COD$ is the change in COD concentration (mg l$^{-1}$) before and after each cycle.

# 3. Results and discussion

## 3.1. COD removal performance

In order to study the influence of influent organic load on the water quality treatment performance of different electrode spacing systems, the carbon source amount was increased after every 28 d of operation of the system, and the average influent COD concentration was $139.88 \pm 5.99$ mg l$^{-1}$ (C/N = 2), $249.88 \pm 34.27$ mg l$^{-1}$ (C/N = 4), $493.63 \pm 6.20$ mg l$^{-1}$ (C/N = 8), $793 \pm 4.87$ mg l$^{-1}$ (C/N = 10), respectively.

The COD change trend in the BRC–MFC system with different electrode spacing was shown in figure 2. The results showed that the COD removal rate of the BRC3 system was higher than other electrode spacing systems in the four stages. The reason was that the cathode area of the BRC4 system was at the top of the filler layer (5 cm from the top). Although it was in aerobic conditions, the

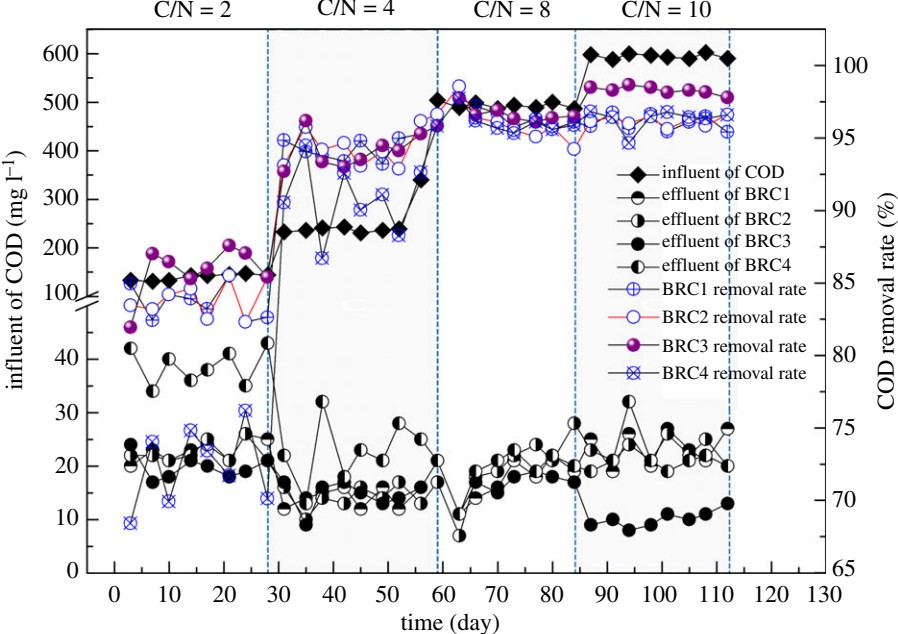

**Figure 2.** COD variation trend of different electrode spacing.

electrode spacing of 40 cm was too large and the electron transfer capacity was limited. The cathode areas of the BRC1 and BRC2 systems were in the lower middle of the filler layer. It was very close to the anode zone, but there was not enough oxygen in the cathode zone, which limited the oxidation reaction of organic matter in the cathode zone. Yu *et al.* [20] also obtained similar results, the COD removal effect of the cathode zone in the middle of the reaction column was more obvious than on the surface.

The study also found that the COD removal rate of different electrode spacing systems increased with the increase of the organic load of the influent, which was the same as the conclusion that the COD removal rate was positively correlated with the organic load [21,22]. Especially in the high organic load (C/N = 10) operation, the COD removal rate between the electrode spacing was significant ($p <$ 0.01), up to 98.49% (BRC3 system). It was higher than the previous research results [23].

## 3.2. $NH_4^+ - N$ removal performance

In order to study the effect of different electrode spacing on ammonia nitrogen removal performance, the system monitored the $NH_4^+ - N$ in the inlet and outlet water, as shown in figure 3. It could be seen that the removal rate of $NH_4^+ - N$ in the BRC3 system was higher than other systems in the four stages. The reason was that $NH_4^+ - N$ was mainly removed by nitrification in an aerobic environment [24], because the cathode areas of the BRC1 and BRC2 systems were under the filler layer, the cathode area was insufficiently oxygen, which led to a decrease in the metabolic rate of nitrifying bacteria [25] and the denitrification efficiency lower than the BRC3 system; although the cathode area of the BRC4 system was under aerobic conditions (the cathode area was 5 cm from the top), the distance between the electrodes was too large (40 cm) and the electron transfer capacity was restricted, which limited the denitrification efficiency.

The study found that with the increase of influent C/N, the $NH_4^+ - N$ removal rates of the four BRC–MFC systems all had a corresponding improvement, especially under the operating condition of C/N = 10, the $NH_4^+ - N$ removal of different electrode spacing systems was significant ($p < 0.01$), the highest removal rate was 97% (BRC3 system), which was higher than the previous research levels [26,27]. On the one hand, the higher $NH_4^+ - N$ removal efficiency might be that the system used activated carbon as an electrode, because the surface of activated carbon was a good medium for attaching microorganisms [28], the functional group of activated carbon would enhance the adhesion of microorganisms on the surface of activated carbon [29], and the amount of microorganisms in the electrode area increased to improve the $NH_4^+ - N$ biodegradability; on the other hand, it might be that under high organic load operation, the carbon source was sufficient, which would help improve

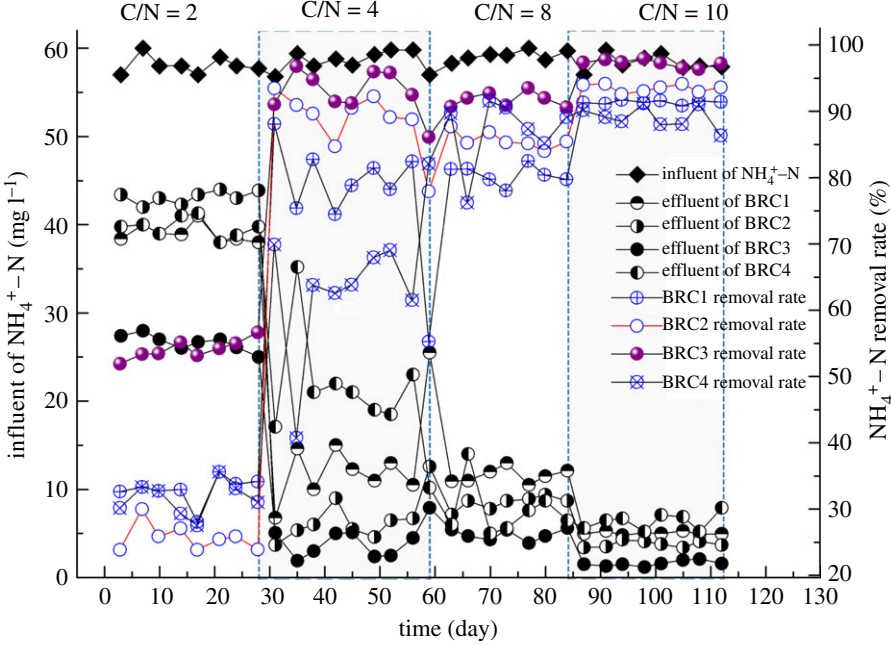

**Figure 3.** $NH_4^+ - N$ variation trend of different electrode spacing.

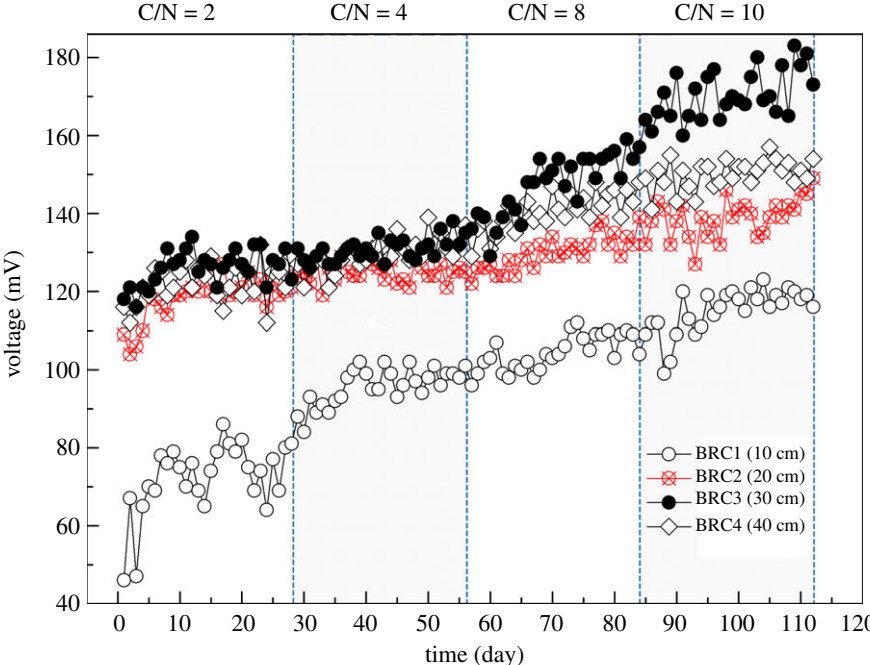

**Figure 4.** Variation trend of output voltage under different electrode spacing.

the system's nitrogen removal efficiency [30]. It could be seen that increasing the influent C/N was an effective measure to improve the $NH_4^+ - N$ removal rate.

## 3.3. Electricity production efficiency

The voltage curve of the BRC–MFC system with different electrode spacing under different water C/N conditions was shown in figure 4. In the same C/N operating cycle, the output voltage at the start-up stage showed an increasing trend, but it was basically stable after 7 d of operation. According to the curve, it could be seen that the electrode spacing had a certain influence on the electricity generation

**Table 2.** Comparison of electricity generation performance with different electrode spacing.

| spacing (cm) | | output voltage (mV) | | | | coulomb efficiency (%) | | | |
|---|---|---|---|---|---|---|---|---|---|
| | | C/N = 2 | C/N = 4 | C/N = 8 | C/N = 10 | C/N = 2 | C/N = 4 | C/N = 8 | C/N = 10 |
| spacing 1 | 10 | 72.21 ± 9.10 | 95.82 ± 4.64 | 104.32 ± 4.43 | 115.18 ± 5.61 | 0.88 | 0.58 | 0.31 | 0.22 |
| spacing 2 | 20 | 118.32 ± 5.32 | 124.82 ± 2.54 | 130 ± 4.29 | 138.93 ± 4.94 | 1.44 | 0.76 | 0.39 | 0.26 |
| spacing 3 | 30 | 126.00 ± 3.54 | 130.79 ± 2.88 | 147.36 ± 7.72 | 170.46 ± 6.17 | 1.50 | 0.85 | 0.44 | 0.31 |
| spacing 4 | 40 | 122.05 ± 4.99 | 129.64 ± 4.11 | 140.57 ± 4.75 | 150.11 ± 3.67 | 1.22 | 0.82 | 0.42 | 0.28 |

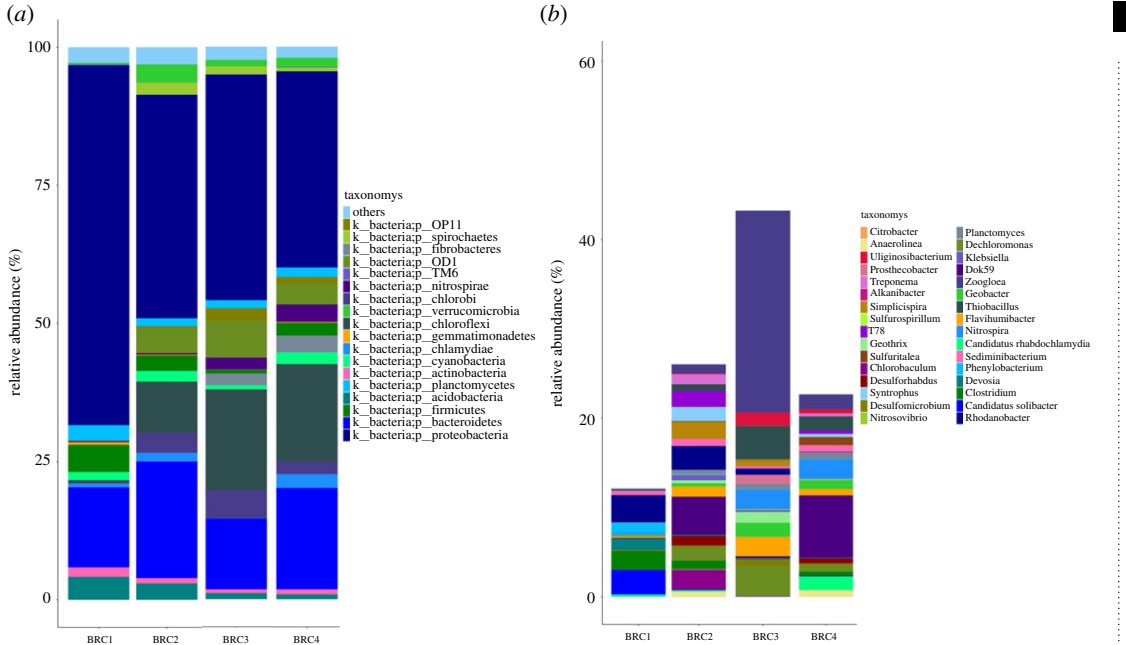

**Figure 5.** Classification of dominant bacterial phylum and genus (relative abundance greater than or equal to 1%). (*a*) Phylum level (*b*) genus level.

efficiency of the BRC–MFC system. Studies had shown that the voltage increased with the decrease of the electrode spacing [31]. However, the conclusions of this study were inconsistent with the conclusions of previous studies [27,31]. This study obtained the lowest voltage output in the BRC1 system, which had the smallest electrode spacing (10 cm). Because the electrode spacing was too small, some of the electrons would be directly used by the microorganisms in the anode area, and the electrons did not pass through the external circuit, thereby it obtained a low voltage output [32], resulting in the deterioration of the electricity production of the BRC1 system. Oon [33] also reached a similar conclusion, obtaining the lowest voltage output in the A1-C system with the smallest electrode spacing (15 cm).

It could be seen from figure 4 that during the entire experiment, the voltage output of the BRC–MFC system with different electrode spacing increased with the increase of C/N. The same conclusion has been reached [34]. Due to the increase in carbon source, microorganisms tended to consume more carbon source (as fuels) to maintain metabolic rate. Therefore, the oxidation of organic matter produced more protons and electrons, which contributed to the increase in voltage output. It could be found that the BRC3 system had the highest output voltage under the operating condition of C/N = 10, with an average output voltage of $170 \pm 6.17$ mV. This might be because the carbon source in the BRC1 system and BRC2 system was completely consumed and degraded by the microorganisms on the upper part of the reaction column. When the sewage reached the cathode area of the system, the nutrients were not enough to support the microbial activity in the corresponding area, resulting in a lower output voltage. It might also be because the electrode spacing of the BRC4 system was too large and the electron transfer capacity was limited, resulting in a lower output voltage [35].

The coulombic efficiency (CE) of the BRC–MFC system with different electrode spacing was shown in table 2. From table 2, the maximum CE value appeared in the BRC3 system. The CE values in this system were all higher than the research results of the CW-MFC systems, such as 0.1–0.36% by Doherty *et al.* [36], 0.2–0.3% by Fang *et al.* [21], and 0.6% by Oon *et al.* [27]. However, compared with other systems, such as Venkata Mohan's research result of 27.03%, this system was still quite low. The phenomenon of low CE in this system showed that only a small part of the oxidized substrate was used as an electron donor for biological electricity generation, and most of the substrate was used for anaerobic digestion [37]. At the same time, Logan also had similar research conclusions that CE might be affected when certain substrates were consumed for processes such as methane production and fermentation instead of cell synthesis by electroactive bacteria.

At the same time, a higher CE value could be obtained under a higher organic load (higher C/N), because as the organic load increased in the anode area, there was sufficient carbon source in the

(a)

(b)

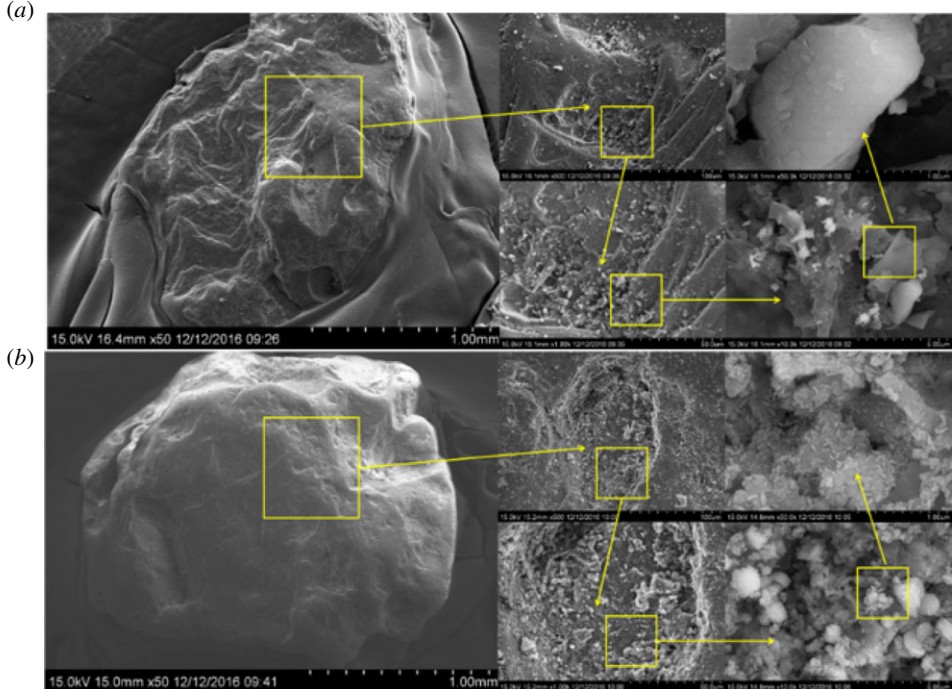

**Figure 6.** SEM images of attachment layer in BRC1 (*a*) and BRC3 (*b*).

**Table 3.** Relative abundance distribution (unit: %).

|  | bacteria phylum of BRC3 | bacteria genera of BRC3 |
| --- | --- | --- |
| Top 10 | *Proteobacteria*, 41 | Dok59, 9 |
|  | *Chloroflexi*, 18 | *Geobacter*, 6 |
|  | *Bacteroidetes*, 13 | *Chlorobaculu*, 3 |
|  | OD1, 7 | *Dechloromonas*, 2 |
|  | *Chlorobi*, 5 | *Devosia*, 2 |
|  | *Fibrobacteres*, 2 | *Anaerolinea*, 1 |
|  | *Nitrospirae*, 2 | *Thiobacillus*, 1 |
|  | OP11, 2 | *Treponema*, 1 |
|  | *Acidobacteria*, 1 | *Uliginosibacterium*, 0.2 |
|  | *Actinobacteria*, 1 | *Syntrophus*, 0.2 |

system to support the growth and activities of microorganisms in the bottom area, so the current output also increased, and led to higher power generation.

## 3.4. Analysis of systemic functional microorganisms

In order to further understand the microbial community structure under different conditions, the samples were analysed at different classification levels. The distribution of each group of samples at the gate level is shown in figure 5, which shows the species with relative abundance greater than 1%, and those with relative quantity less than 1% and unclassified are classified as others.

Many scholars have detected and proved that Proteobacteria, Firmicutes, *Acidobacteria* and *Bacteroidetes* have electrochemical activity [38], *Citrobacter* [39], *Geobacter* [23], *Clostridium* [40] and *Geothrix* [41] are dominant genera of electricity producing bacteria. These were found in the bioelectrical enhancement system (figure 5).

It can be seen from figure 5 and table 3 that the relative abundances of *Proteobacteria* and *Bacteroidetes* in the BRC3 system gradually increased with time, reaching 41% and 13%. The relative abundance of

Dok59 and *Geobacter* was 9% and 6% in all genera. This indicates that the improvement of denitrification performance may be due to the participation of both: *Geobacter* may participate in the process of enhancing the system electronic supply, and Dok59 may participate in the process of nitrogen removal. The difference in microbial community structure may be caused by the composition of the packed sediment fuel cell and its degradation products [42]. In the BRC system, the content of *Rhodanobacter* remained relatively stable over time, with a content of 3%. It can be seen that there is no obvious change of microbial community structure in the system without bioelectricity enhancement intervention, and anaerobic denitrifying bacteria are the main species.

This can also be confirmed by SEM characterization of particles in BRC1 and BRC3. As shown in figure 6, the particle surface uniformity of BRC3 is significantly higher than that of BRC1, and the number of microbial communities attached to BRC3 is also significantly higher than that of BRC1.

# 4. Conclusion

The main purpose of this research was to explore the optimal electrode spacing of the BRC–MFC system for wastewater treatment. The study found that when the electrode spacing was 30 cm, it had significant water treatment capacity and high output voltage.

In addition, the results showed that sufficient carbon sources would also improve the water treatment capacity and electricity generation efficiency.

A proper electrode spacing could effectively improve the water treatment capacity of the BRC–MFC system. Nevertheless, to make the water treatment capacity of the BRC–MFC system achieve the best effect by changing the electrode spacing, further experiments are needed.

Data accessibility. Our data are deposited at Dryad: https://doi.org/10.5061/dryad.nvx0k6dr1.
Authors' contributions. W.Y-J: Conceptualization, methodology, writing—original draft; C.T.-J.: investigation, formal analysis, visualization; L.J.-S.: Data curation; S.Y.-M.: supervision; W.Z.-Y.: conceptualization, methodology, writing—review and editing.
Competing interests. We declare we have no competing interests
Funding. We received no funding for this study.
Acknowledgements. This research was co-funded by the National Natural Science Foundation of China (grant no. 41967043), Gansu Province Natural Science Foundation (grant no. 20JR5RA461) and Industrial support programme of higher education of Gansu province (2020C-40).

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
