## [Peer Review File · Royal Society Open Science]

Review History

RSOS-202024.R0 (Original submission)

Review form: Reviewer 1

Is the manuscript scientifically sound in its present form?

Yes

Are the interpretations and conclusions justified by the results?

Yes

Is the language acceptable?

Yes

Do you have any ethical concerns with this paper?

Yes

Have you any concerns about statistical analyses in this paper?

Yes

Recommendation?

Accept with minor revision (please list in comments)

Comments to the Author(s)

The manuscript reports the Experimental study on the influence of electrode spacing on the performance of bioretention cell coupled with MFC. Overall the experiments are performed well and also manuscript was well written, I recommend this research manuscript is appropriate to publish in this journal with minor revision mentioned below.

Avoid using words such as we, he, she, them, their, etc.

Language is good, but, proof reading by a native speak would avoid the minor errors.

Please concrete the keywords and make it formally and academically.

About the Introduction section to make the reading more clear and smooth it should be organized according to the following items: i) present state of the art; ii) literature review; iii) motivation and objective of the study proposed; iv) innovative contribution in terms of methodology developed.

Authors are suggested to read the following articles and cite them in the appropriate places.

Introduction must be revised to enhance readability i.e. I would also like to see following references in the revised version

<https://doi.org/10.1016/j.fuel.2019.05.109>

<https://doi.org/10.1016/j.biortech.2020.123110>

<https://doi.org/10.1002/er.5734>

<https://doi.org/10.1016/j.fuel.2019.115682>

<https://doi.org/10.1016/j.fuel.2020.118119>

Typographical errors are present throughout the manuscript. Authors are required to pay keen attention to this

In the materials and methods, divide them into nice sub-sections. Provide the details of all the equipment's, instruments used, their model number, company of manufacture, country, etc.

In the introduction section, write the novelty of the work and the problem statement clearly.

70% of the references should be from 2018, 2019 and 2020. Kindly do a careful literature review.

The results and discussion section is ONLY to write your results and facilitate scientific / technical discussions. Provide mechanism based reactions and refer to important/recent literatures in the results and discussions

Conclusions (<100 words) should be in line with the specific objectives of your work. Do not repeat the results and the methodology here

Delete unwanted old references. Refer to references from the years 2017, 2018 and 2019. Figures number must rearrange in this manuscript.

Authors must compare more results with previous publications mainly in characterization parts (Results and discussion)

Only few previous publications in discussions. Why not authors compare the current results with previous publications.

Authors are required to re-write the conclusion.

Authors must check the manuscript carefully before submit to journal. Because still manuscript having grammar mistake and some words joined with some other words.

Authors must show good results in abstract section to enhance the readers.

Authors must use any one format throughout the manuscript for example minutes must be as "min"; "ml" must be as "mL" and hours must be as "h". Throughout the manuscript

Please avoid reference overkill/run-on - do not use more than 3 references per sentence. If you need to use more, make sure you state the key relevant idea of each reference.

Many sentences could be seen with our enough evidences or justifications - you appropriate references to justify your arguments.

Additionally, please pay attention to revise/check the following:

1- English language throughout the manuscript. Please ensure to have a proof reading to your manuscript.

2- Similarity index shall not exceed 15% with no more than 1% from any source.

3- Citation of references with latest publications on the topic.

4- Manuscript shall have an adequate number of Figures and Tables.

5- Figures must be in high quality format. Please ensure also to submit a high quality Graphical Abstract representing a summary of your study.

Review form: Reviewer 2 (Monika Sogani)

Is the manuscript scientifically sound in its present form?

Yes

Are the interpretations and conclusions justified by the results?

Yes

Is the language acceptable?

Yes

Do you have any ethical concerns with this paper?

No

Have you any concerns about statistical analyses in this paper?

No

Recommendation?

Accept as is

Comments to the Author(s)

This study indicated ways for improving the performance of the BRC-MFC system by varying the electrode spacing on the performance of the system. The data presented seems sufficient to corroborate results. May be considered for publication in its present form.

Decision letter (RSOS-202024.R0)

Dear Dr Wang:

Title: Experimental study on the influence of electrode spacing on the performance of bioretention cell coupled with MFC

Manuscript ID: RSOS-202024

Thank you for submitting the above manuscript to Royal Society Open Science. On behalf of the Editors and the Royal Society of Chemistry, I am pleased to inform you that your manuscript will be accepted for publication in Royal Society Open Science subject to minor revision in accordance with the referee suggestions. Please find the reviewers' comments at the end of this email. I apologise it has taken longer than usual to be able to send you this decision.

The reviewers and handling editors have recommended publication, but also suggest some minor revisions to your manuscript. Therefore, I invite you to respond to the comments and revise your manuscript.

Because the schedule for publication is very tight, it is a condition of publication that you submit the revised version of your manuscript before 09-May-2021. Please note that the revision deadline will expire at 00.00am on this date. If you do not think you will be able to meet this date please let me know immediately.

Kind regards,
Dr Laura Smith
Publishing Editor, Journals

On behalf of the Subject Editor Professor Anthony Stace and the Associate Editor Dr Darren Walsh.

RSC Associate Editor:
Comments to the Author:
(There are no comments.)

RSC Subject Editor:
Comments to the Author:
(There are no comments.)

Reviewer comments to Author:

Reviewer: 1

Comments to the Author(s)

The manuscript reports the Experimental study on the influence of electrode spacing on the performance of bioretention cell coupled with MFC. Overall the experiments are performed well and also manuscript was well written, I recommend this research manuscript is appropriate to publish in this journal with minor revision mentioned below.

Avoid using words such as we, he, she, them, their, etc.

Language is good, but, proof reading by a native speak would avoid the minor errors.

Please concrete the keywords and make it formally and academically.

About the Introduction section to make the reading more clear and smooth it should be organized according to the following items: i) present state of the art; ii) literature review; iii) motivation and objective of the study proposed; iv) innovative contribution in terms of methodology developed.

Authors are suggested to read the following articles and cite them in the appropriate places.

Introduction must be revised to enhance readability i.e. I would also like to see following references in the revised version

<https://doi.org/10.1016/j.fuel.2019.05.109>

<https://doi.org/10.1016/j.biortech.2020.123110>

<https://doi.org/10.1002/er.5734>

<https://doi.org/10.1016/j.fuel.2019.115682>

<https://doi.org/10.1016/j.fuel.2020.118119>

Typographical errors are present throughout the manuscript. Authors are required to pay keen attention to this

In the materials and methods, divide them into nice sub-sections. Provide the details of all the equipment's, instruments used, their model number, company of manufacture, country, etc.

In the introduction section, write the novelty of the work and the problem statement clearly.

70% of the references should be from 2018, 2019 and 2020. Kindly do a careful literature review.

The results and discussion section is ONLY to write your results and facilitate scientific / technical discussions. Provide mechanism based reactions and refer to important/recent literatures in the results and discussions

Conclusions (<100 words) should be in line with the specific objectives of your work. Do not repeat the results and the methodology here

Delete unwanted old references. Refer to references from the years 2017, 2018 and 2019. Figures number must rearrange in this manuscript.

Authors must compare more results with previous publications mainly in characterization parts (Results and discussion)

Only few previous publications in discussions. Why not authors compare the current results with previous publications.

Authors are required to re-write the conclusion.

Authors must check the manuscript carefully before submit to journal. Because still manuscript having grammar mistake and some words joined with some other words.

Authors must show good results in abstract section to enhance the readers.

Authors must use any one format throughout the manuscript for example minutes must be as "min"; "ml" must be as "mL" and hours must be as "h". Throughout the manuscript

Please avoid reference overkill/run-on - do not use more than 3 references per sentence. If you need to use more, make sure you state the key relevant idea of each reference.

Many sentences could be seen with our enough evidences or justifications - you appropriate references to justify your arguments.

Additionally, please pay attention to revise/check the following:

1- English language throughout the manuscript. Please ensure to have a proof reading to your manuscript.

2- Similarity index shall not exceed 15% with no more than 1% from any source.

3- Citation of references with latest publications on the topic.

4- Manuscript shall have an adequate number of Figures and Tables.

5- Figures must be in high quality format. Please ensure also to submit a high quality Graphical Abstract representing a summary of your study.

Reviewer: 2

Comments to the Author(s)

This study indicated ways for improving the performance of the BRC-MFC system by varying the electrode spacing on the performance of the system. The data presented seems sufficient to corroborate results. May be considered for publication in its present form.

Author's Response to Decision Letter for (RSOS-202024.R0)

See Appendix A.

RSOS-202024.R1 (Revision)

Review form: Reviewer 1

Is the manuscript scientifically sound in its present form?

Yes

Are the interpretations and conclusions justified by the results?

Yes

Is the language acceptable?

Yes

Do you have any ethical concerns with this paper?

No

Have you any concerns about statistical analyses in this paper?

Yes

Recommendation?

Accept as is

Comments to the Author(s)

Accepted for publications

Review form: Reviewer 2 (Monika Sogani)

Is the manuscript scientifically sound in its present form?

Yes

Are the interpretations and conclusions justified by the results?

Yes

Is the language acceptable?

Yes

Do you have any ethical concerns with this paper?

No

Have you any concerns about statistical analyses in this paper?

No

Recommendation?

Accept as is

Comments to the Author(s)

All the reviewers comments have been answered by the authors satisfactorily. The manuscript may be considered for publication.

Decision letter (RSOS-202024.R1)

Dear Dr Wang:

Title: Experimental study on the influence of electrode spacing on the performance of bioretention cell coupled with MFC

Manuscript ID: RSOS-202024.R1

It is a pleasure to accept your manuscript in its current form for publication in Royal Society Open Science. The chemistry content of Royal Society Open Science is published in collaboration with the Royal Society of Chemistry.

On behalf of the Subject Editor Professor Anthony Stace and the Associate Editor Dr Darren Walsh.

RSC Associate Editor:
Comments to the Author:
(There are no comments.)

RSC Subject Editor:
Comments to the Author:
(There are no comments.)

Reviewer(s)' Comments to Author:
Reviewer: 1
Comments to the Author(s)
Accepted for publications

Reviewer: 2
Comments to the Author(s)
All the reviewers comments have been answered by the authors satisfactorily. The manuscript may be considered for publication.

Appendix A

07-May-2021

Dr Laura Smith
Editor
Royal Society Open Science

Subject: Submission of revised manuscript for your kind consideration (Manuscript ID: RSOS-202024).

Dear Editor,

Thank you very much for considering our manuscript for review and providing valuable comments by imminent reviewers. We have checked and revised our manuscript very carefully according to reviewer's comments. We have also checked the entire manuscript for language errors and corrected it carefully. All the changes made in our revised manuscript have been marked in YELLOW color and detailed response to reviewer's comments have been provided as given bellow.

Kindly consider our revised manuscript for further evaluation and publication in Royal Society Open Science.

Your's sincerely,

Zhaoyang Wang
Lanzhou University
Lanzhou, China 730050

Response to Reviewer's comments:

Reviewer: 1

The manuscript reports the Experimental study on the influence of electrode spacing on the performance of bioretention cell coupled with MFC. Overall the experiments are performed well and also manuscript was well written, I recommend this research manuscript is appropriate to publish in this journal with minor revision mentioned below.

Response: Thank you very much for your positive and encouraging comments on our work. We sincerely appreciate your kind comments and suggestions. Please find bellow the response to each comments.

1) Avoid using words such as we, he, she, them, their, etc.

Response: Thank you very much for your kind suggestion. We have made changes in the revised manuscript.

2) Language is good, but, proof reading by a native speak would avoid the minor errors.

Response: Thank you very much for your encouraging and constructive suggestion. We have got native speakers proof reading. All the changes made in our revised manuscript have been marked in YELLOW color.

3) Please concrete the keywords and make it formally and academically.

Response: Thank you very much for your kind suggestion. We have added the keywords information in the revised manuscript.

“However, Bioretention Cells (BRCs) are similar to constructed wetlands, which is a kind of ecological treatment system, using physical, chemical and biological effects to achieve removal effect, imitating the natural environment, and have good wastewater treatment effects without the need for energy input.”

4) About the Introduction section to make the reading more clear and smooth it should be organized according to the following items: i) present state of the art; ii) literature review; iii) motivation and objective of the study proposed; iv) innovative contribution in terms of methodology developed.

Response: Authors sincerely thank reviewer for this very important suggestion. We have revised Introduction in the revised manuscript.

“Regarding water pollution, the world is facing two important challenges, one is compound pollutant pollution, and the other is energy shortage. In order to solve these two problems at the same time, a sustainable wastewater treatment technology has gradually become a research hotspot.

Traditional wastewater treatment plants rely on the combined effects of physics, chemistry and biology. This process requires mechanical equipment and a large amount of energy input to achieve the removal of pollutants. However, Bioretention Cells (BRCs) are similar to constructed wetlands, which is a kind of ecological treatment system, using physical, chemical and biological effects to achieve removal effect, imitating the natural environment, and have good wastewater treatment effects without the need for energy input. Studies had shown that BRC had a significant removal effect on pollutants such as TSS, COD and heavy metals. Due to its small size, simple structure, good treatment effect, low cost, and strong sustainability, it has become a potential wastewater treatment technology. At present, the researches on BRC mainly focuses on the operation efficiency of facilities and the optimization of design parameters (Such as matrix type, gradation, hydraulic load, height of submerged layer, etc).

The biogebattery reflected a natural phenomenon that occurred at the oxidation-reduction interface of the earth's surface, it used electrons generated by microorganisms oxidizing organic carbon, sulfide and other electron donors in the anaerobic area were transmitted to the aerobic area through a "long distance" through the extracellular mediator, and then underwent a reduction reaction with electron acceptor such as oxygen. However, the operating principle of microbial fuel cells (MFCs) is similar to this. In recent years, as an emerging technology, it has great potential for simultaneous wastewater treatment and electricity generation, and it has attracted widespread attention from scholars. MFC use wastewater pollutants as fuels

and convert these into electrical energy through electrochemically active microbial degradation compounds, making MFC a sustainable technology. In order to effectively play the MFC function, the anode area must be kept in an anaerobic state, while the oxygen in the cathode area combines with protons and electrons to form a circuit. The reactions that generally occur in MFC can be summarized by the following equation (Eqs.(1)–(3)):

There had been some early studies on independent BRC systems and MFC systems. However, the coupling of BRC and MFC is still in the initial stage of research. In this study, BRC and MFC were coupled to form a BRC-MFC system, in this system, the entire reaction column had both anaerobic and aerobic conditions. When the organic matter in the anode area was oxidized, it would produce the electrons (e-) and protons (H+), then moved to the cathode, and the external circuit transferred the electrons from the anode to the cathode through the insulated copper wire to generate current. By studying the effect of electrode spacing on the performance of BRC-MFC, on the one hand, this study could test the feasibility of the coupling system of BRC and MFC to treat wastewater and generate electricity at the same time, which laid the foundation for future research, and on the other hand to explore the influence of electrode spacing on the water treatment capacity of the BRC-MFC system, and to obtain the best electrode spacing, which provided reference for future practical application.”

5) Authors are suggested to read the following articles and cite them in the appropriate places. Introduction must be revised to enhance readability i.e. I would also like to see following references in the revised version

<https://doi.org/10.1016/j.fuel.2019.05.109>

<https://doi.org/10.1016/j.biortech.2020.123110>

<https://doi.org/10.1002/er.5734>

<https://doi.org/10.1016/j.fuel.2019.115682>

<https://doi.org/10.1016/j.fuel.2020.118119>

Typographical errors are present throughout the manuscript.

Response: Thank you very much for your kind suggestion. We have added these references and revised Introduction in the revised manuscript.

For typographical errors, we have corrected and changed in the entire revised manuscript.

6) Authors are required to pay keen attention to this

In the materials and methods, divide them into nice sub-sections.

Response: Thank you very much for your kind suggestion. We have add sub-sections in the revised manuscript.

7) Provide the details of all the equipment's, instruments used, their model number, company of manufacture, country, etc.

Response: Thank you very much for your kind suggestion. We have added the details in the revised manuscript.

“A multi-channel peristaltic pump (WT600-2J, Longer, China)

an HACH DR2800 (DR2800, Hach, USA)

a spectrophotometer UV-1800 (UV-1800, Shimadzu Corp., Japan)

a Midi LOGGER GL820 data acquisition instrument (GL820, Japan Graphic Technology Co., Ltd., Japan)

a VC890D multimeter (VC890D, Shenzhen Yisheng Shengli Technology Co., Ltd., China)”

8) In the introduction section, write the novelty of the work and the problem statement clearly.

Response: Authors sincerely thank reviewer for this very important suggestion. We have corrected the novelty of the work and the problem statement in the entire revised manuscript.

“There had been some early studies on independent BRC systems and MFC systems. However, the coupling of BRC and MFC is still in the initial stage of research. In this study, BRC and MFC were coupled to form a BRC-MFC system, in this system, the entire reaction column had both anaerobic and aerobic conditions. When the organic matter in the anode area was oxidized, it would produce the electrons (e-) and protons (H+), then moved to the cathode, and the external circuit transferred the electrons from the anode to the cathode through the insulated copper wire to generate current. By studying the effect of electrode spacing on the performance of BRC-MFC, on the one hand, this study could test the feasibility of the coupling system of BRC and MFC to treat wastewater and generate electricity at the same time, which laid the foundation for future research, and on the other hand to explore the influence of electrode spacing on the water treatment capacity of the BRC-MFC system, and to obtain the best electrode spacing, which provided reference for future practical application.”

9) 70% of the references should be from 2018, 2019 and 2020. Kindly do a careful literature review.

Response: Thank you very much for your kind suggestion. We have updated some of the references and changed in the revised manuscript.

10) The results and discussion section is ONLY to write your results and facilitate scientific / technical discussions. Provide mechanism based reactions and refer to important/recent literatures in the results and discussions

Conclusions (<100 words) should be in line with the specific objectives of your work.

Do not repeat the results and the methodology here

Delete unwanted old references. Refer to references from the years 2017, 2018 and 2019. Figures manuscript must rearrange in this manuscript.

Response: Thank you very much for your kind suggestions. We have added the related mechanism-based reactions and references and changed in the revised manuscript.

For Conclusions, we have corrected and changed in the entire revised manuscript.

“The main purpose of this research was to explore the optimal electrode spacing of the BRC-MFC system for wastewater treatment. The study found that when the electrode spacing was 30 cm, it had significant water treatment capacity and high output voltage. In addition, the results showed that sufficient carbon sources would also improve the water treatment capacity and electricity generation efficiency. A proper electrode spacing could effectively improve the water treatment capacity of the BRC-MFC system. Nevertheless, to make the water treatment capacity of the BRC-MFC system achieve the best effect by changing the electrode spacing, further experiments are needed.”

We have deleted some old references and added some new references. The figures manuscript of manuscript has been arranged.

11) Authors must compare more results with previous publications mainly in characterization parts (Results and discussion). Only few previous publications in discussions. Why not authors compare the current results with previous publications.

Response: Thank you very much for your kind query. We have added some previous publications in Results and discussion and changed in the revised manuscript.

“[22] Chen K Q, Sun W X, Fu S F, et al. Treatment of simulated textile printing and dyeing wastewater by anaerobic baffled reactor [J]. Water treatment technology, 2019, 45 (03): 44-48.”

“[23]Thulasinathan B , Ebenezer J O , Bora A , et al. Bioelectricity generation and analysis of anode biofilm metabolites from septic tank wastewater in microbial fuel cells[J]. International Journal of Energy Research, 2020(3).”

“[34] Zang W S, Zhou X G, Li H Z, et al. The effect of pH value and carbon nitrogen ratio on the denitrification and phosphorus removal effect of microbial fuel cell [J]. Journal of irrigation and drainage, 2019, 038 (002): 49-55.”

12) Authors are required to re-write the conclusion.

Response: Thank you very much for your kind suggestion. We have re-written the conclusion and changed in the revised manuscript.

“The main purpose of this research was to explore the optimal electrode spacing of the BRC-MFC system for wastewater treatment. The study found that when the electrode spacing was 30 cm, it had significant water treatment capacity and high output voltage.

In addition, the results showed that sufficient carbon sources would also improve the water treatment capacity and electricity generation efficiency.

A proper electrode spacing could effectively improve the water treatment capacity of the BRC-MFC system. Nevertheless, to make the water treatment capacity of the BRC-MFC system achieve the best effect by changing the electrode spacing, further experiments are needed.”

13) Authors must check the manuscript carefully before submit to journal. Because still manuscript having grammar mistake and some words joined with some other words.

Response: Thank you very much for your kind suggestion. We have corrected mistakes and changed in the entire revised manuscript.

14) Authors must show good results in abstract section to enhance the readers.

Response: Thank you very much for your kind suggestion. We have corrected in the revised manuscript.

15) Authors must use any one format throughout the manuscript for example minutes must be as “min”; “ml” must be as “mL” and hours must be as “h”. Throughout the manuscript

Please avoid reference overkill/run-on - do not use more than 3 references per sentence. If you need to use more, make sure you state the key relevant idea of each reference.

Response: Thank you very much for your kind suggestion. We have corrected format and changed in the revised manuscript. For references, we have also changed in the revised manuscript.

16) Many sentences could be seen with our enough evidences or justifications - you appropriate references to justify your arguments.

Response: Authors sincerely thank Editor for valuable suggestion. We have replaced some long sentences with references and changed in the revised manuscript.

17) Additionally, please pay attention to revise/check the following:

1- English language throughout the manuscript. Please ensure to have a proof reading to your manuscript.

2- Similarity index shall not exceed 15% with no more than 1% from any source.

3- Citation of references with latest publications on the topic.

4- Manuscript shall have an adequate number of Figures and Tables.

5- Figures must be in high quality format. Please ensure also to submit a high quality Graphical Abstract representing a summary of your study.

Response: Thank you very much for your kind suggestions. We have a proof reading to all manuscript and changed in the revised manuscript. We ensure the similarity index not exceed 15% with no more than 1% from any source. We have also updated some of the references and changed in the revised manuscript. And we have an adequate number of Figures and Tables in manuscript. The Figures and Graphical Abstract are also of high quality format.

Reviewer: 2

This study indicated ways for improving the performance of the BRC-MFC system by varying the electrode spacing on the performance of the system. The data presented

seems sufficient to corroborate results. May be considered for publication in its present form.

Response: Thank you very much for your approval on our work. We are also eager to be accepted for publication in Royal Society Open Science.